# Providing environmental enrichments can reduce subclinical spondylolisthesis prevalence without affecting performance in broiler chickens

**Marconi Italo Lourenço da Silva**[1,2☯]*, **Ibiara Correia de Lima Almeida Paz**[1☯], **Andressa Silva Jacinto**[1☯], **Marcos Antonio Nascimento Filho**[1☯], **Ana Beatriz Santos de Oliveira**[1☯], **Ingrid Grazieli Althman dos Santos**[1☯], **Francine dos Santos Mota**[1☯], **Fabiana Ribeiro Caldara**[3☯], **Leonie Jacobs**[2‡]

**1** Department of Animal Production and Preventive Veterinary Medicine, School of Veterinary Medicine and Animal Sciences (FMVZ), São Paulo State University "Júlio de Mesquita Filho" (UNESP), Botucatu, São Paulo, Brazil, **2** School of Animal Sciences, Virginia Tech, Blacksburg, VA, United States of America, **3** Department of Animal Production, College of Agrarian Sciences, Federal University of Grande Dourados, Dourados, Mato Grosso do Sul, Brazil

☯ These authors contributed equally to this work.
‡ LJ is a joint senior author on this work.
* marconi.italo@unesp.br

**Data Availability Statement:** The datasets generated and analyzed during the current study

## Abstract

Environmental enrichment can increase the occurrence of natural behavior and improve leg health and other animal welfare outcomes in broiler chickens. This study aimed to assess the effects of three environmental enrichments, specifically hay bales, step platforms, and laser lights, on subclinical spondylolisthesis prevalence, productivity, behavior, and gait of broiler chickens (*Gallus gallus domesticus*). Twenty-four hundred day-old male Ross® AP95 chicks from a commercial hatchery were used in a completely randomized design with four treatments and four replicate pens per treatment. Pens contained either a Control (C) treatment, an environment similar to a commercial broiler chicken system without environmental enrichments, or an environment with either additional hay bales (HB), additional step platforms (SP), or additional laser lights (LL). Performance, yield, behavior (frequencies), gait score, and subclinical spondylolisthesis prevalences were assessed. When raised with SP or LL access, fewer chickens had subclinical spondylolisthesis than chickens without enrichments (C) or with HB access. Chickens with access to SP exhibited higher wing yield and less abdominal fat than animals from the C group. Chickens from the LL and HB treatments explored more and rested less frequently than animals from the C and SP treatments. As chickens aged, they became less active, exploring less and increasing resting and comfort behaviors. Treatments did not affect gait. Gait was not associated with subclinical spondylolisthesis prevalence. Environmental enrichments benefitted chicken health (subclinical spondylolisthesis) and behavior (exploration) without negative consequences for performance and yield.

are available in http://dx.doi.org/10.17632/xp62w777ks.1.

**Funding:** LOURENÇO-SILVA, MI; ALMEIDA PAZ, ICL Latin American Poultry Association (ALA) https://www.avicolatina.com/ LOURENÇO-SILVA, MI Coordination for the Improvement of Higher Education Personnel (CAPES) https://www.gov.br/capes The funders had no role in study design, data collection and analysis, decision to publish, or preparation of the manuscript.

**Competing interests:** The authors have declared that no competing interests exist.

## Introduction

Locomotory activity is strongly associated with broiler chicken welfare; many behavioral patterns that depend on locomotion, such as exploration, seeking food, water, shelter, and escaping predators, are negatively affected by the poor walking ability in fast-growing broiler chickens [1]. Rapid muscle growth and exacerbated development of the *Pectoralis major* muscle in fast-growing broiler chickens change the chickens' center of gravity, altering the chickens' posture and load on the skeleton compared to slower-growing strains, leading to skeletal-biomechanical imbalances, in turn affecting walking ability and resulting in locomotor disorders [2, 3].

Spondylolisthesis, also known as 'kinky back', is a deformity that affects the thoracic vertebral arch of chickens. It occurs in the sixth vertebra (T6), leading to spinal cord compression and locomotor difficulties and, in turn possibly resulting in paralysis of the pelvic limbs in severe cases [4, 5]. This condition can often subclinically affect chickens between 3 and 6 weeks of age before it evolves into clinical cases [6]. Subclinical spondylolisthesis can affect 15 to 47% of the flock, while clinical spondylolisthesis can affect 2% of the flock [7–9]. Chickens with subclinical spondylolisthesis do not show symptoms, while broilers with clinical spondylolisthesis are lame and will sit with extended feet, show an imbalance, and fall on their side when attempting to stand [8, 10–13]. A positive correlation between gait score and subclinical spondylolisthesis suggests this is a health and animal welfare concern [7].

The main cause of this condition is a genetic predisposition. Indigenous chickens (slow-growing) show a good balance, better gait, and no subclinical spondylolisthesis compared to fast-growing chickens [5, 7, 11, 12]. However, recent studies did not confirm this [14, 15]. It is probable that a combination of both genetic predisposition and environment (housing, nutrition) can lead to high prevalences of subclinical spondylolisthesis [4]. According to [7], tendons and bones in recent strains of broiler chickens are too weak to support the chickens' body weight due to the immaturity of the musculoskeletal system. When providing diets diluted with wood sawdust fiber in the first week of life, prevalence of subclinical spondylolisthesis was reduced but daily gain was also stunted [16].

Locomotor disorders can to some extent be prevented by increasing animal activity, as locomotion can positively impact skeletal development [17, 18]. Positive activity such as exploratory behavior and locomotion can be stimulated through environmental enrichments, such as straw or hay bales, platforms, and moving laser lights [17–20].

Environmental enrichments increase the complexity of broiler chicken environments and enhance the expression of natural behaviors, i.e., foraging, perching, and playing [19, 21]. Foraging is a highly motivated behavior associated with exploration and the appetitive phase of feeding behavior. Chickens peck and scratch the ground searching for food while collecting environmental information [22]. Perching is natural resting behavior; chickens seek an elevated place that allows them to perform surveillance behavior against predators [23, 24]. Playing is another natural and social behavior; chickens run, jump, and interact with inanimate objects and conspecifics in a non-aggressive way [25, 26].

Potentially beneficial enrichments are straw or hay bales, platforms, and moving laser lights. Hay bales can increase chicken activity by stimulating pecking, foraging, preening, worm running (play), and can provide a barrier for resting [27–29]. Platforms provide animals an elevated area for resting with fewer physical challenges compared to perches [17, 23]. In addition, platforms allow chickens to perform natural locomotory behavior, such as jumping and walking up and down a ramp, which may improve chickens' musculoskeletal strength and coordination, and in turn prevent skeletal-biomechanical imbalances [19, 30]. Laser light enrichments have been used in previous research and seem a valuable resource to increase

broilers' locomotion [19, 31–33]. Laser lights can simulate the presence of an insect, stimulating chickens to approach the stimulus [34] and increasing locomotion during early life [19, 34], which in turn may benefit the development of their musculoskeletal system.

The effects of environmental enrichments on performance and yield depend on the type of resource used [35]. Studies assessing peat, bales, elevated platforms, perches, mirrors, balls, dust bathing substrates, and pecking objects (hanging metal chains) at 5 or 6 weeks of age for fast-growing broilers have reported no effects of these enrichments on final body weight, feed conversion ratio, mortality, or the percentage of animals rejected in the slaughterhouse [30, 36–38]. In contrast, access to perches and ramps reduced body weights and feed intake compared to the control group, with no difference in feed conversion ratios [39, 40]. Average daily gain and body weights improved in slow-growing chickens at 9 weeks of age when they had access to bales and perches, but feed conversion ratio and mortality were worsened compared to the control group [41]. Due to these inconsistencies, more research is needed to understand the impact of environmental enrichments on productivity.

Potential effects of environmental enrichments on subclinical spondylolisthesis have not previously been examined. Therefore, this study aimed to assess the effects of three environmental enrichments, specifically hay bales, step platforms, and laser lights, on subclinical spondylolisthesis prevalence, performance, behavior, and gait of broiler chickens. We hypothesized that each environmental enrichment would increase exploratory behaviors in broilers compared to an unenriched control. In turn, we hypothesized that enrichments would reduce the prevalence of subclinical spondylolisthesis (platforms>bales>laser lights) and improve gait scores without affecting productivity compared to an unenriched control.

## Materials and methods

The experiment was carried out at the School of Veterinary Medicine and Animal Sciences (FMVZ) at São Paulo State University, Botucatu, SP, Brazil (22˚ 49' 07" S and 48˚ 24' 40" W). The experimental protocol was approved by the Animal Use Ethics Committee of FMVZ (number 0045/2020 CEUA).

### Chickens, facilities, and management

Twenty-four hundred day-old male Ross® AP95 chicks from a commercial hatchery were used. The trial was carried out in a climate-controlled poultry barn featuring negative pressure ventilation. The barn contained sixteen pens (4 x 3 m) provided with 10 cm new wood shavings as bedding, three semi-automatic feeders (one feeder for 50 chickens), and a nipple drinker line (one nipple for 10 chickens). The litter was turned on days 17, 24, and 31. Each pen contained 150 chickens and a targeted maximum stocking density of 39 kg/m$^2$ [42]. Pens contained heat lamps in the first 10 days. House temperature was gradually decreased from 32˚C on day 1 to 21˚C on day 28 and remained 21˚C until day 42. The chickens were maintained on an artificial lighting program of 24L:0D in the first 10 days due to the heat lamps and 16L:8D until the end of the trial. The corn- and soybean-meal-based diets were adapted from [43] and met the nutritional requirements in three rearing phases: starter (1–21 days, 24% CP and 3,000 kcal ME/kg), grower (22–35 days, 22.5% CP and 3,150 kcal ME/kg), and finisher (36–42 days, 19% CP and 3,250 kcal ME/kg). Both feed and water were provided *ad libitum*.

### Experimental design

The trial followed a completely randomized design with four treatments and four replicate pens per treatment. Pens contained either a Control (C) treatment (Fig 1A), which consisted of an environment similar to commercial broiler chicken husbandry without environmental

enrichments, or an environment with either additional hay bales (HB), step platforms (SP), or laser lights (LL). At 42 days of age, the stocking densities (mean±SE) calculated for each treatment were: C = 36.5±0.9 kg/m$^2$, HB = 36.0±0.9 kg/m$^2$, SP = 35.8±1.7 kg/m$^2$, and LL = 37.6 ±0.6 kg/m$^2$.

## Environmental enrichment

All resources were introduced on the first day and remained available until day 42. One hay bale (75 x 42 x 30 cm, alfalfa hay) per HB pen (150 chickens) was provided and replaced on day 35. The bale was placed between the drinker line and the barn wall (Fig 1B). Step platforms (60 × 60 × 7 cm for the lower base and 20 × 20 × 7 cm for the upper base) were made from MDF boards and one was provided in every SP pen (Fig 1C). When the litter was turned, the platforms were scraped to remove excreta. Step platforms were placed between the drinker line and the barn wall. One laser light projector per LL pen was placed at 1.5m height (Mini Stage Lighting XX-027, Spooboola, China). The projector emitted approximately 36 green and red laser lights simultaneously with wavelengths of 532nm (50mW) and 650nm (100mW; Fig 1D). These downwards projected lights moved across an area of approximately 12 m$^2$ at a slow and steady pace. Projectors were turned on twice a day for 15 minutes, at 09:00 and 15:00 hours.

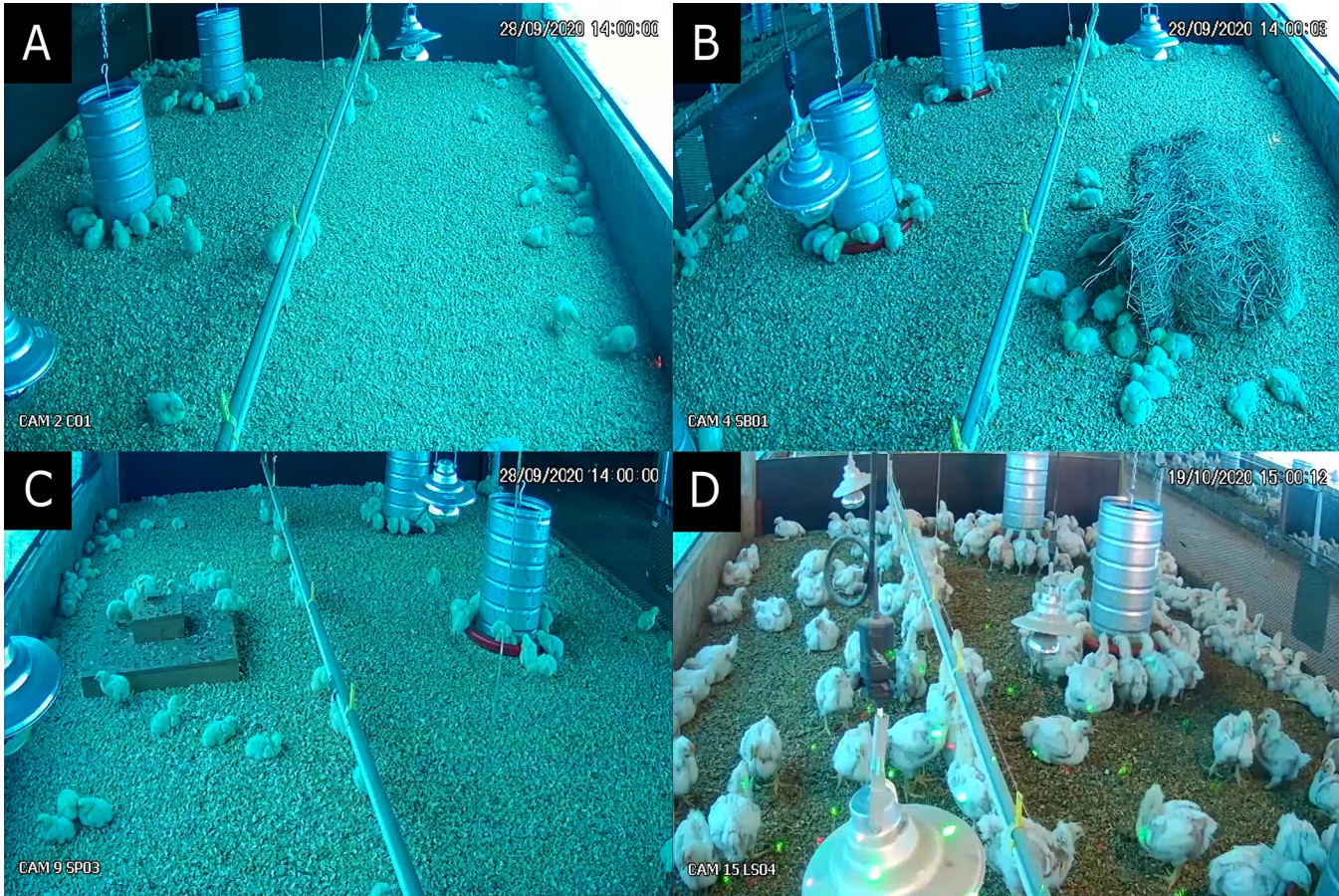

**Fig 1. Broiler chickens housed in four environments.** (A) Control, (B) Hay bale, (C) Step platform, and (D) Laser lights.

**Table 1. Experimental ethogram of recorded broiler chicken behaviors, based on [44].**

| Behavior | Operational definition |
| --- | --- |
| **Consummatory** | |
| Eating | Chicken holds its head above the feeding trough or the surrounding area and actively taking in food |
| Drinking | Chicken is actively taking in water by pecking at nipple drinkers |
| **Resting** | |
| Sitting/resting | Chicken sits on the litter while the head rests on the ground or upright; eyes may be open or closed |
| Sitting/resting by the bale (HB treatment) | Chicken sits in immediate proximity to bale (within 10 cm) while the head rests on the ground or upright; eyes may be open or closed |
| Rest on top of bale (HB treatment) | Chicken stands or lies on top of a straw bale |
| Rest on step platform (SP treatment) | Chicken stands or lies on top of a step platform |
| **Exploratory** | |
| Locomotion | Moving using legs in a continuous forward motion (walking or running), chicken may be flapping wings. |
| Foraging | Pecking and/or scratching at the flooring substrate |
| Play | After approach of another chicken at high speed, the chicken stops and faces the other briefly, without making physical contact. |
| Pecking at hay bales (HB treatment) | Chicken uses beak to manipulate hay in the bale |
| Chasing and/or pecking at light (LL treatment) | Chicken approaches and/or pecks the light emitted by the laser projector |
| **Comfort** | |
| Preening | All behavior patterns associated with cleaning and maintenance of its body surface using the beak; the chicken may stand or lie |
| Dust bathing | Vertical wing shakes, interacting with flooring substrate, performing side-rubs, and intermittent ground pecking with beak |

## Measurements

**Behavior.** Sixteen high-resolution video cameras (Intelbras, São José, SC, Brazil) were installed at 1.5m height (angled down) to record behavior in each pen (total of 16 pens). Videos were recorded on days 6 (week 1), 13 (week 2), 20 (week 3), 27 (week 4), 34 (week 5), and 41 (week 6). Human disturbance was limited on recording days, and birds were only disturbed for health checks. Frequencies of all selected behavioral occurrences were recorded at pen-level by a single trained observer using 1-minute continuous scan sampling observation with 2 minutes inter-sampling intervals for two 15-minute time periods (starting at 09:00 and 15:00 hours). This resulted in 5 scans per time period and a targeted total of 960 behavioral entries (5 scans × 2 time points × 16 pens × 6 weeks). Behaviors were coded following the ethogram adapted from [44] (Table 1). After recording, behaviors were classified into four categories: consummatory, resting, exploratory, and comfort. Then, the frequencies of each behavioral category were calculated. In addition, the frequency of behaviors associated with the environmental enrichments was assessed individually.

**Gait score.** A trained observer assessed all chickens' gait scores (GS) at 21, 28, 35, and 42 days of age. A three-point scale was applied to classify the gait of the chickens according to [45]: score 0 (GS 0) was attributed to healthy chickens that exhibited no abnormality when walking, score 1 (GS 1) was attributed to chickens that exhibited difficulty in walking in a way that was easily identifiable through observation, and score 2 (GS 2) was attributed to chickens exhibiting severe issues walking. Chickens presenting GS 2 were euthanized as a humane

endpoint, and their scores were noted. Before gait assessment at 42 days of age, 48 chickens of each treatment (12 per pen) were arbitrarily selected and leg banded for subclinical spondylolisthesis prevalence assessment. Then, gait assessment proceeded in all chickens.

**Subclinical spondylolisthesis prevalence.** At 43 days of age, the previous leg banded chickens were fasted for 8 hours. Fasting is a common procedure prior to processing to allow emptying of the gastro-intestinal tract and is required by the Brazilian Ministry of Agriculture for food safety considerations (SDA/MAPA Ordinance No. 365, of 16 July 2021). The chickens were weighed and then stunned using an electric stunner (Fluxo UFX 7, Chapecó, SC, Brazil). The chickens were exsanguinated via a cut to the carotid artery and jugular vein. After slaughter, the feet, head, and neck were removed. The chickens' backs were frozen for 48 hours and then sawed sagittally to assess subclinical spondylolisthesis by visualizing the cervical spine between the 6th and the 7th vertebrae macroscopically (Fig 2). When the vertebrae were found on their normal axis and without compressing the bone marrow, the score was 0 (absence of subclinical spondylolisthesis). When the vertebrae compressed the bone marrow, the score was 1 (presence of subclinical spondylolisthesis) [7, 11]. The observer was blinded to the treatments.

**Performance and yield.** All chickens and feed were weighed by pen on days 21, 35, and 42 to determine average body weight gain, feed intake, and feed conversion ratio. Mortality and culls were recorded daily. After the slaughter at 43 days, the feet, head, and neck were removed and warm carcasses were weighed to calculate carcass yield (% of live weight). Carcass parts were weighed separately to calculate yields (% of carcass weight) of the breast with skin and bones, wings, legs, back, breast fillet, boneless legs, and abdominal fat.

### Statistical analysis

The data were analyzed using SAS Studio 3.8 (SAS Inst. Inc., Cary, NC, USA). The variance homogeneities were assessed by Levene's test and data residuals' normality was verified by the Shapiro-Wilk test. Performance and yield data were subjected to ANOVA using the GLM procedure, followed by Tukey's multiple comparison test, and assigned significance when $P < 0.05$. Behavioral data were subjected to ANOVA using the MIXED procedure, followed by Tukey's multiple comparison test and assigned significance when $P < 0.05$ with treatment ($n = 4$), weeks ($n = 6$), and their interactions ($n = 24$) as fixed effects, pen as a random effect, and time period (9:00 and 15:00 h) as a repeated factor using the "variance components" covariance structure. A generalized linear mixed model was applied for gait score data using a multinomial (ordered) distribution; gait score was the response variable, treatment ($n = 4$), age ($n = 4$), and their interactions ($n = 16$) were fixed effects, and pen was a random effect. Subclinical spondylolisthesis prevalence data were subjected to a generalized linear mixed model using a binary distribution with treatment ($n = 4$) as a fixed effect and pen as a random effect. The correlation between gait score and subclinical spondylolisthesis prevalence was assessed using Spearman's correlation analysis with the CORR procedure, considering a significance of $P < 0.05$.

## Results

### Behavior

Treatment effects for resting ($F_{3,168} = 3.22$, $P = 0.024$, Fig 3A) and exploratory ($F_{3,168} = 13.62$, $P < 0.001$, Fig 3B) behavior frequencies were found. Chickens raised in the control treatment rested more frequently than chickens in the hay bale ($P = 0.018$) or laser light ($P = 0.037$) treatments but at similar frequency compared to chickens in the step platform treatment ($P = 0.788$). The laser light treatment stimulated more frequent exploratory behavior than hay

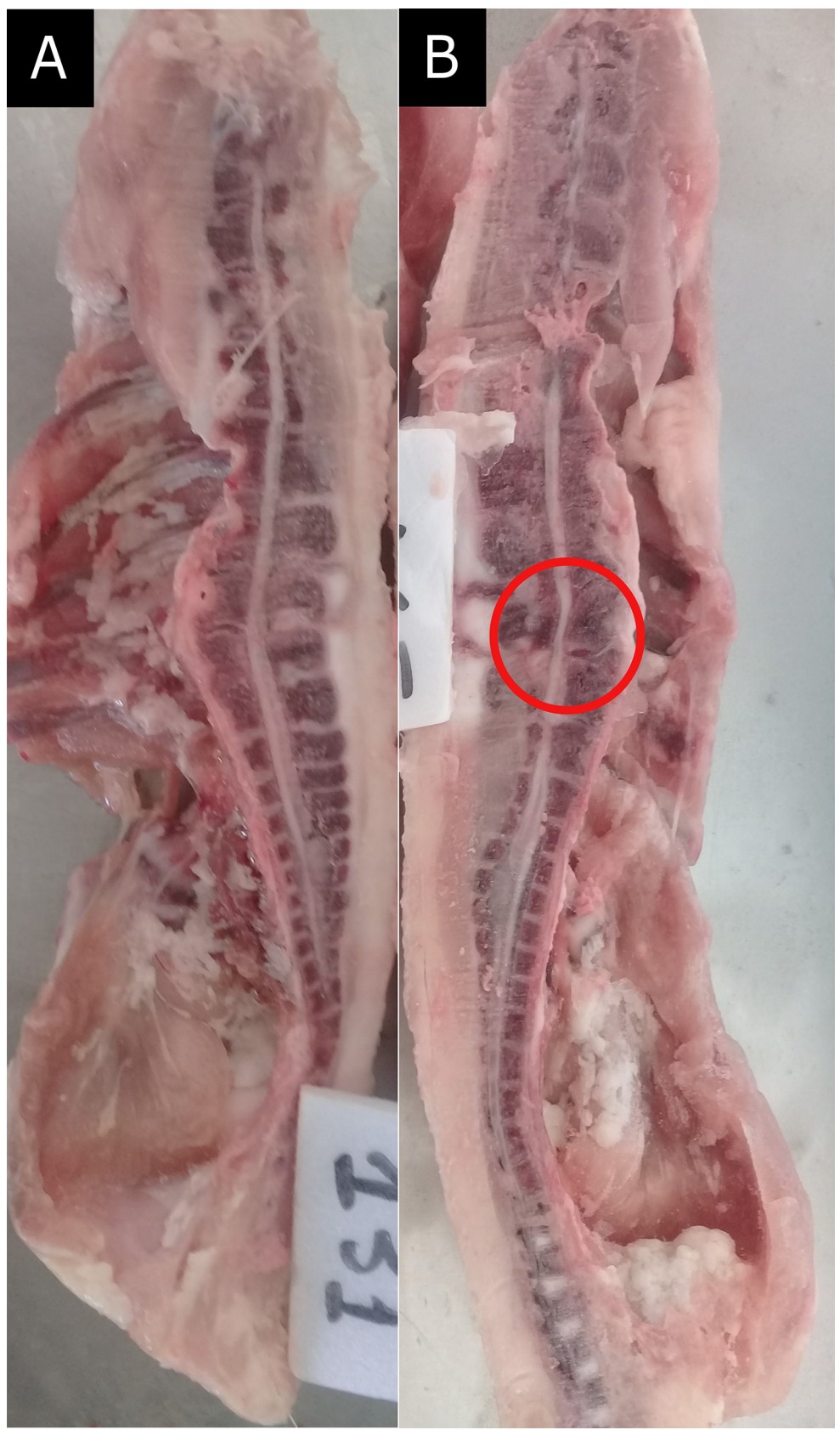

**Fig 2. Presence or not of subclinical spondylolisthesis in broiler chickens' backs at 43 days of age.** (A) Score 0: vertebrae found on their normal axis without compressing the bone marrow, (B) Score 1: vertebrae compressing the bone marrow.

bale (P = 0.027), step platform (P < 0.001), and control (P < 0.001) treatments. Chickens in the hay bale treatment explored more frequently than those in step platform (P = 0.002) and control (P = 0.002) treatments. Exploratory behavior frequencies for step platform and control treatments did not differ (P = 0.989). No treatment effects were found on comfort ($F_{3,168}$ = 2.16, P = 0.095, Fig 3C) or consummatory ($F_{3,168}$ = 0.72, P = 0.539, Fig 3D) behavior frequencies. No interactions between treatment and age were found for any of the four behavioral categories (P > 0.144).

Chickens rested more ($F_{5,168}$ = 2.38, P = 0.041, Fig 4A) and showed more comfort behavior ($F_{5,164}$ = 34.18, P < 0.001, Fig 4B) but showed less frequent exploratory behavior ($F_{5,168}$ = 32.62, P < 0.001, Fig 4C) with age. Similarly, enrichments were used less frequently with age, with decreased frequencies of pecking hay bales ($F_{5,42}$ = 3.56, P = 0.009, Fig 5), chasing and/or pecking laser lights ($F_{5,42}$ = 10.78, P < 0.001, Fig 6), and resting near hay bales ($F_{5,42}$ = 5.80,

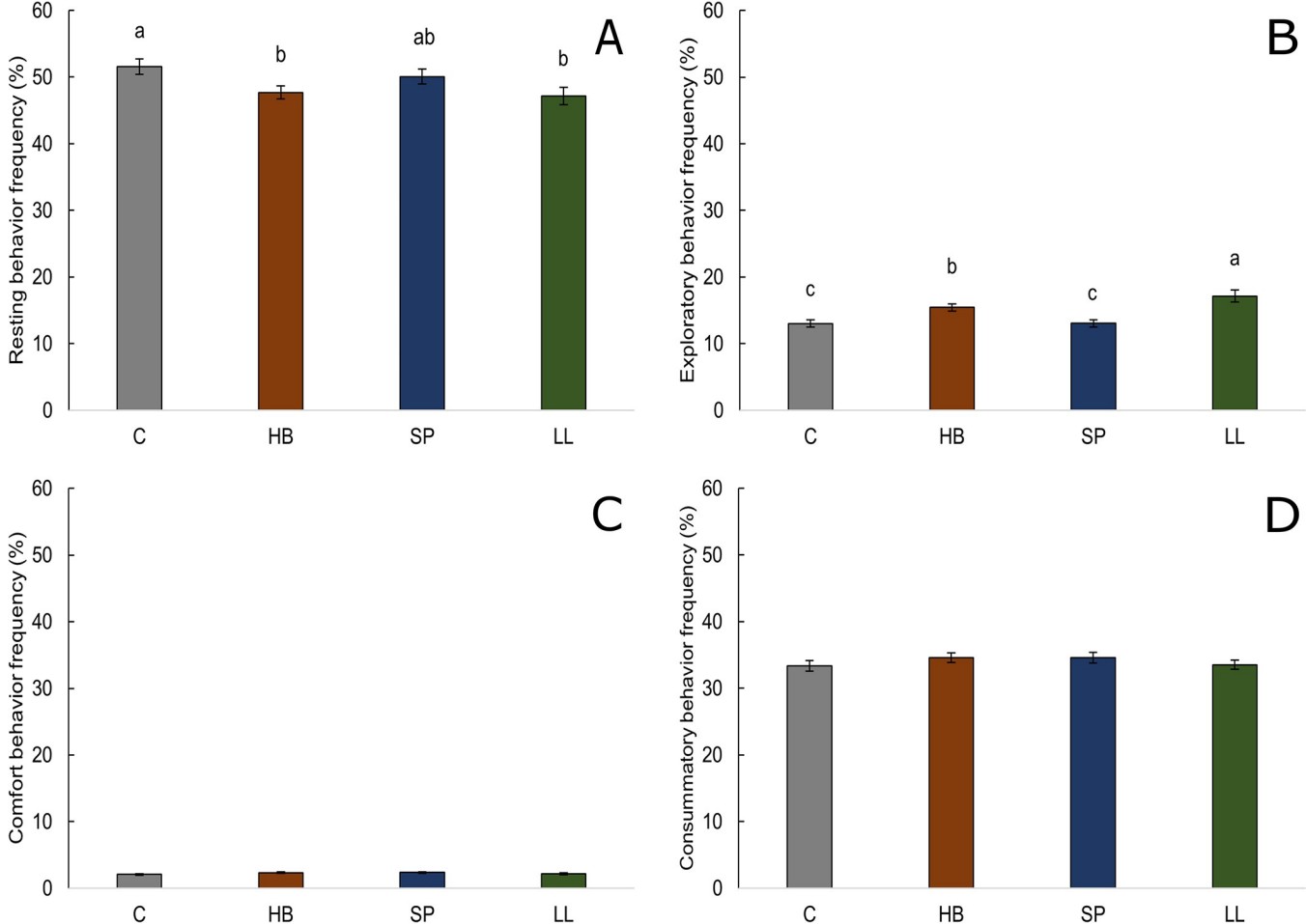

**Fig 3. Frequency of observations (mean % ± SEM) of each behavioral category by treatment (C = control, HB = hay bales, SP = step platforms, LL = laser lights) across 2 time periods (09:00 and 15:00 h), n = 192 observations.** Frequencies of (A) resting behavior, (B) exploratory behavior, (C) comfort behavior, and (D) consummatory behavior. Means within behavioral category without a common superscript ([a–c]) differed at P < 0.05.

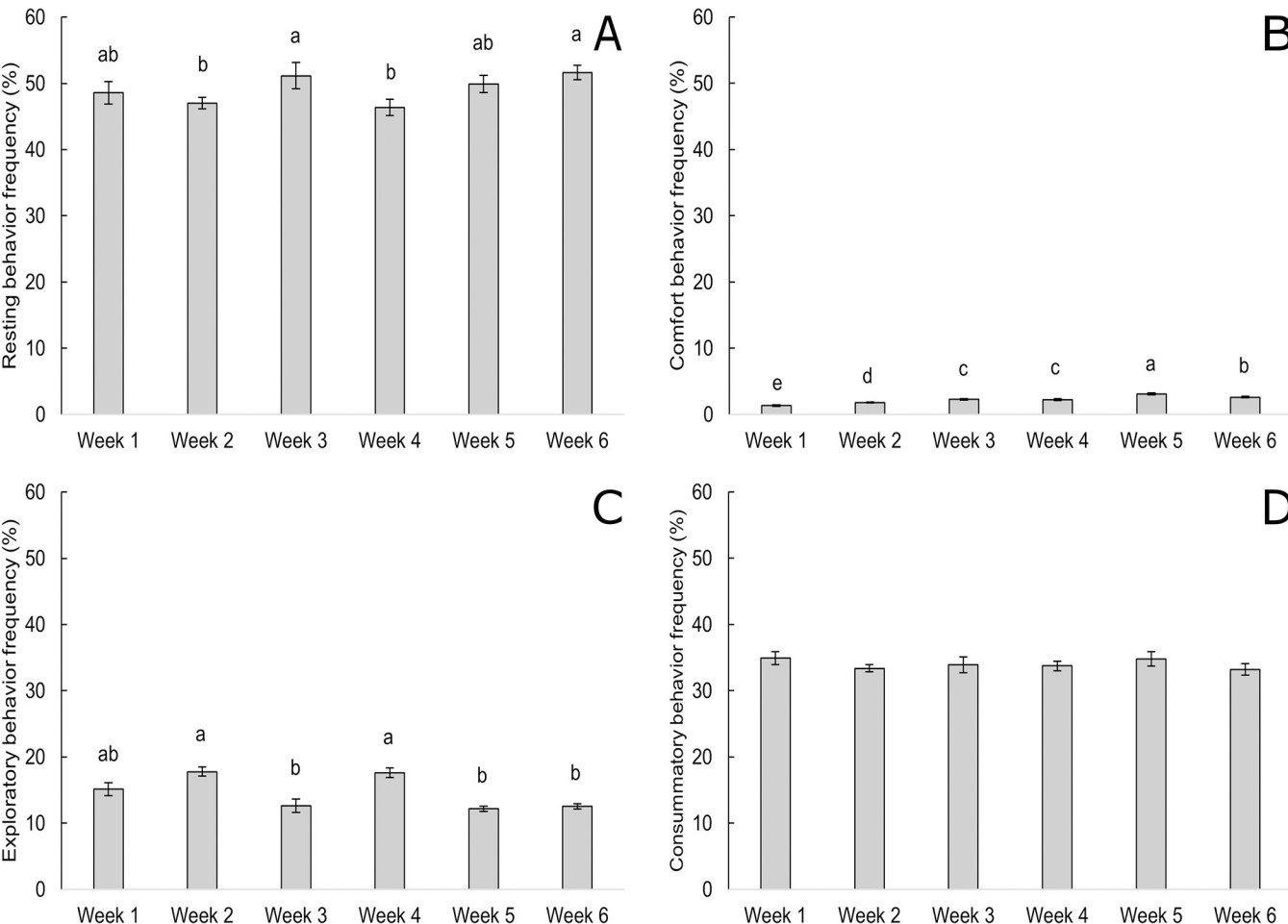

**Fig 4. Frequency of observations (mean % ± SEM) of each behavioral category by chicken age (in weeks) across two time periods (09:00 and 15:00 h), n = 192 observations.** Frequencies of (A) resting behavior, (B) comfort behavior, (C) exploratory behavior, and (D) consummatory behavior. Means within a behavioral category without a common superscript ([a–e]) differed at P < 0.05.

P < 0.001). Increased frequencies of resting on top of hay bales were observed with advancing age ($F_{5,42}$ = 7.13, P < 0.001). Consummatory behavior ($F_{5,168}$ = 0.57, P = 0.719, Fig 4D) and use of step platforms ($F_{5,42}$ = 1.53, P = 0.201) were not affected by age.

## Gait score

Gait scores were affected by age ($F_{3,135}$ = 37.98, P < 0.001, Fig 7). Chickens showed better gait at 21 days than at 28 (P = 0.004), 35 (P < 0.001) and 42 days (P < 0.001), at 28 days than at 35 (P = 0.002) and 42 days (P < 0.001), and at 35 days than at 42 days of age (P < 0.001). No interaction between treatment and age was found ($F_{9,135}$ = 0.24, P = 0.989). Gait was not affected by treatments ($F_{3,135}$ = 1.37, P = 0.251). Gait was not associated with subclinical spondylolisthesis prevalence (r = 0.010, P = 0.888).

## Subclinical spondylolisthesis prevalence

Subclinical spondylolisthesis prevalence was affected by treatments ($F_{3,45}$ = 5.16, P = 0.002, Fig 8). Chickens from the control treatment showed a higher prevalence than chickens from the

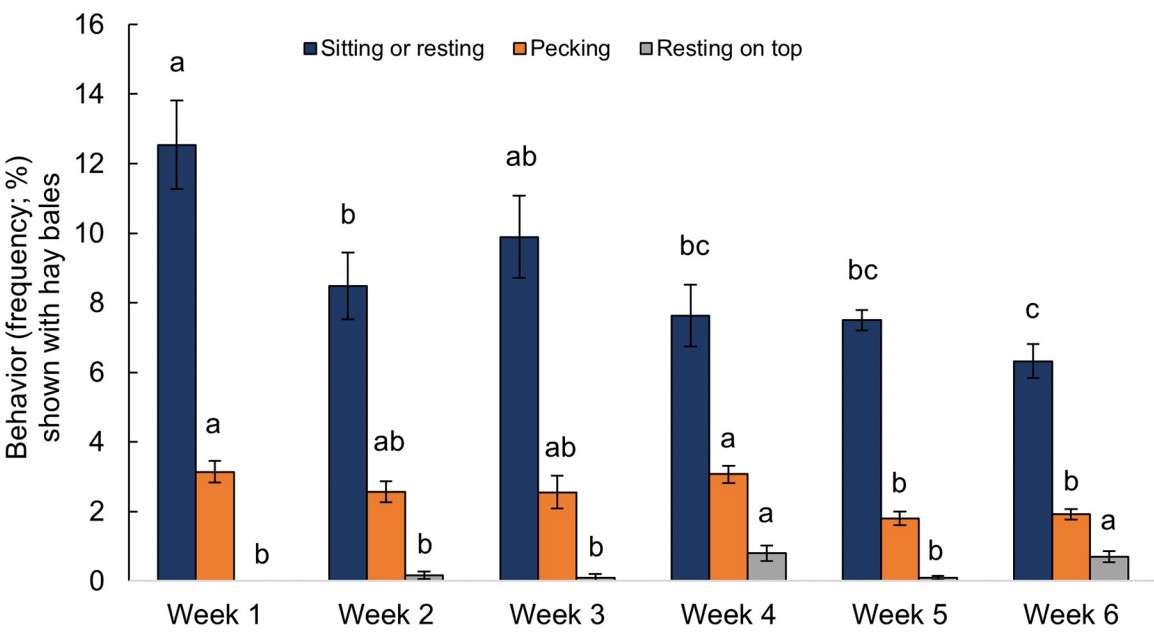

**Fig 5. Frequency of observations (mean % ± SEM) of behaviors associated with the provided hay bale by age (in weeks) across two time periods (09:00 and 15:00 h).** Sitting or resting by bale (n = 24), Pecking (n = 24), and Resting on top (n = 24). Frequencies without a common superscript ([a–c]) differed at P < 0.05.

laser light (P = 0.044) and step platform (P = 0.004) treatments. Chickens from the hay bale treatment had a higher prevalence than chickens from the step platform treatment (P = 0.035). Subclinical spondylolisthesis prevalences did not differ for control and hay bale (P = 0.847), hay bale and laser light (P = 0.245), and laser light and step platform (P = 0.784) treatments.

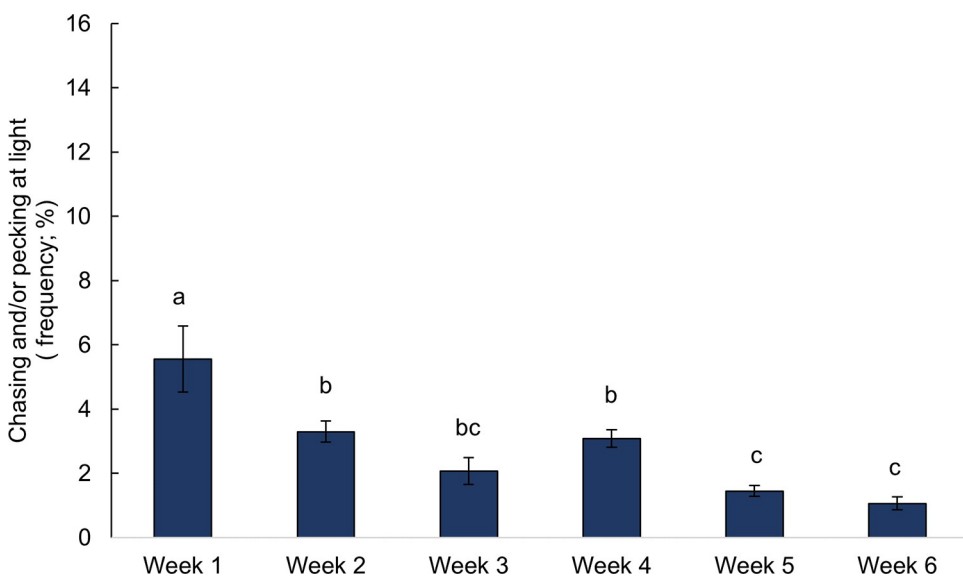

**Fig 6. Frequency of observations (mean % ± SEM) of behaviors associated with the provided laser lights by age (in weeks) across two time periods (09:00 and 15:00 h), n = 24.** Frequencies without a common superscript ([a–c]) differed at P < 0.05.

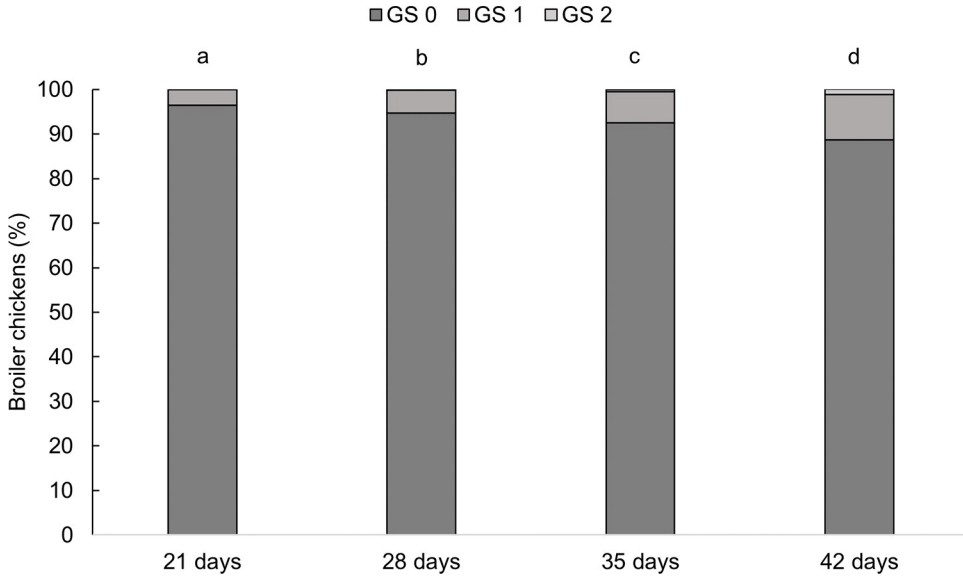

**Fig 7. Frequencies (%) of broiler chickens by gait score (GS) at 21, 28, 35, and 42 days of age, n = 16.** GS 0: Healthy chickens that exhibited no abnormality when walking, GS 1: Chickens that exhibited difficulty walking in a way that was easily identifiable through observation, and GS 2: Chickens exhibiting severe issues walking [45]. Ages without an uncommon superscript ([a–d]) differed at P < 0.05.

## Performance and yield

Live performance parameters were not impacted by treatments in any of the three rearing phases (P > 0.120, Table 2). Chickens raised with access to step platforms had higher wing yield (P = 0.038) and lower abdominal fat yield (P = 0.048) than chickens raised in the control

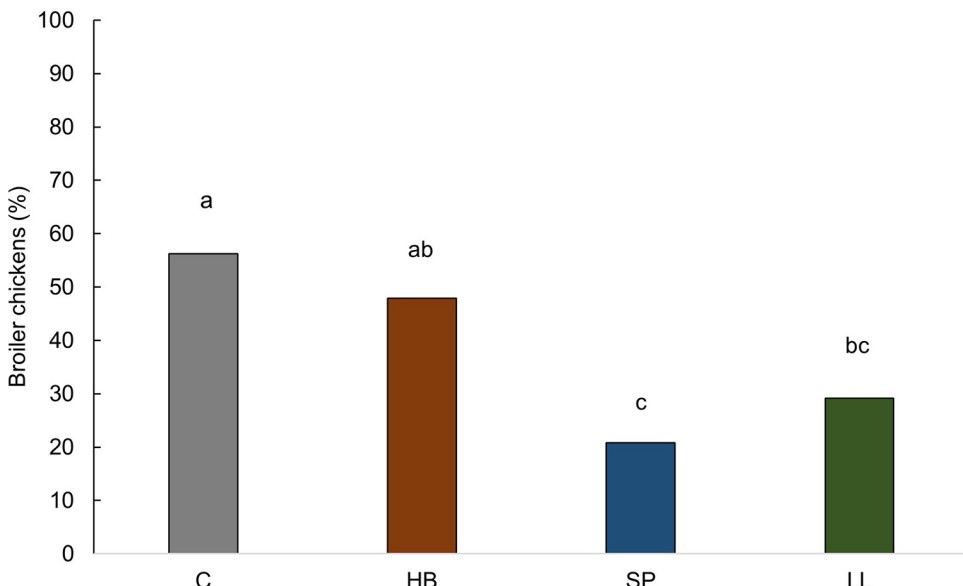

**Fig 8. Proportion (%) of broiler chickens with subclinical spondylolisthesis score 1 (vertebrae compressed the bone marrow) at 43 days of age, n = 192.** C = control, HB = hay bales, SP = step platforms, LL = laser lights. Bars without a common superscript ([a–c]) differed at P < 0.05.

**Table 2. Performance (mean ± SEM) of broiler chickens between days 1–21, days 1–35, and days 1–42 of rearing, n = 16.**

| Performance | Treatments[1] | | | | SEM | ANOVA |
|---|---|---|---|---|---|---|
| | C | HB | SP | LL | | |
| **1–21 days** | | | | | | |
| Feed intake (g/chicken/period) | 1,188 | 1,199 | 1,216 | 1,211 | 7.40 | $F_{3,15} = 0.68, P = 0.583$ |
| Weight gain (g/chicken/period) | 920 | 932 | 951 | 951 | 7.52 | $F_{3,15} = 1.04, P = 0.412$ |
| Feed conversion ratio | 1.29 | 1.29 | 1.28 | 1.27 | 0.01 | $F_{3,15} = 0.52, P = 0.680$ |
| Mortality (%) | 1.34 | 2.18 | 1.01 | 1.34 | 0.19 | $F_{3,15} = 2.28, P = 0.132$ |
| Culls (%) | 1.01 | 0.67 | 0.67 | 0.67 | 0.17 | $F_{3,15} = 0.20, P = 0.894$ |
| **1–35 days** | | | | | | |
| Feed intake (g/chicken/period) | 3,391 | 3,377 | 3,425 | 3,421 | 21.74 | $F_{3,15} = 0.25, P = 0.860$ |
| Weight gain (g/chicken/period) | 2,201 | 2,179 | 2,180 | 2,251 | 27.86 | $F_{3,15} = 0.32, P = 0.811$ |
| Feed conversion ratio | 1.54 | 1.55 | 1.57 | 1.52 | 0.01 | $F_{3,15} = 0.73, P = 0.556$ |
| Mortality (%) | 1.34 | 2.68 | 1.51 | 1.34 | 0.24 | $F_{3,15} = 2.39, P = 0.120$ |
| Culls (%) | 1.17 | 0.84 | 1.51 | 1.34 | 0.18 | $F_{3,15} = 0.61, P = 0.619$ |
| **1–42 days** | | | | | | |
| Feed intake (g/chicken/period) | 5,012 | 4,939 | 5,013 | 5,087 | 42.03 | $F_{3,15} = 0.46, P = 0.715$ |
| Weight gain (g/chicken/period) | 2,999 | 2,967 | 2,949 | 3,093 | 42.04 | $F_{3,15} = 0.53, P = 0.669$ |
| Feed conversion ratio | 1.67 | 1.67 | 1.70 | 1.64 | 0.01 | $F_{3,15} = 0.71, P = 0.562$ |
| Mortality (%) | 2.18 | 3.35 | 2.18 | 2.18 | 0.26 | $F_{3,15} = 1.30, P = 0.319$ |
| Culls (%) | 2.52 | 2.01 | 2.52 | 2.85 | 0.28 | $F_{3,15} = 0.33, P = 0.801$ |

[1] C = control, HB = hay bales, SP = step platforms, LL = laser lights.

group (Table 3). Chickens from the laser light treatment had a lower back yield than the control treatment (P = 0.031). No other differences in yield were found (P > 0.107).

## Discussion

This is the first study to assess the association between environmental enrichments and subclinical spondylolisthesis prevalences. Here we assessed three enrichments that allow broiler chickens to perform natural behavior, reduce subclinical spondylolisthesis prevalence, and maintain performance and yield. The current study showed that access to step platforms and

**Table 3. Yield (mean % ± SEM) of carcass, breast, wings, legs, back, breast fillet, boneless legs, and abdominal fat of broiler chickens slaughtered at day 43 of age, n = 16.**

| Yield (%) | Treatments[1] | | | | SEM | ANOVA |
|---|---|---|---|---|---|---|
| | C | HB | SP | LL | | |
| Carcass[2] | 75.79 | 75.93 | 75.92 | 76.13 | 0.14 | $F_{3,15} = 0.27, P = 0.850$ |
| Breast[3] | 40.78 | 40.82 | 41.03 | 41.69 | 0.15 | $F_{3,15} = 1.96, P = 0.121$ |
| Wings[3] | 10.01 [b] | 10.12 [ab] | 10.28 [a] | 10.08 [ab] | 0.04 | $F_{3,15} = 4.41, P = 0.038$ |
| Legs[3] | 30.69 | 31.08 | 31.14 | 30.73 | 0.10 | $F_{3,15} = 1.24, P = 0.295$ |
| Back[3] | 17.46 [a] | 17.29 [ab] | 17.06 [ab] | 16.73 [b] | 0.09 | $F_{3,15} = 3.02, P = 0.031$ |
| Breast fillet[3] | 31.87 | 31.97 | 32.71 | 32.63 | 0.15 | $F_{3,15} = 2.06, P = 0.107$ |
| Boneless legs[3] | 24.70 | 24.89 | 25.29 | 24.84 | 0.11 | $F_{3,15} = 1.32, P = 0.270$ |
| Abdominal fat[3] | 1.43 [a] | 1.35 [ab] | 1.29 [b] | 1.37 [ab] | 0.03 | $F_{3,15} = 3.99, P = 0.048$ |

[1] C = control, HB = hay bales, SP = step platforms, LL = laser lights. [2] Percentage of carcass yield was calculated based on live weight. [3] Part yields were calculated based on carcass weight. Row means without a common superscript ([a-b]) differed at P < 0.05.

laser lights reduced subclinical spondylolisthesis prevalence compared to a control, which partially aligns with our hypothesis. The prevalence of subclinical spondylolisthesis in fast-growing broiler chickens was previously associated with their fast breast muscle growth, causing unstable equilibrium and in turn leading to postural distress and pressure on the locomotor system [2, 7, 15, 46, 47]. We argue that adding resources to the chickens' environments increases exploratory behaviors and, consequently, locomotion and exercise, which can alleviate pressure on the locomotor system and reduce the prevalence of subclinical spondylolisthesis.

The impact of step platforms and laser lights can be explained by the daily and regular use of step platforms, providing means for exercise, such as walking up, down, and jumping off the platforms, and the increase in exploratory behavior in laser light treatment, such as chasing and/or pecking at the light. This daily exercise likely strengthened their locomotor system, improving their musculoskeletal development [17, 18], thus preventing the skeletal-biomechanical imbalance caused by exacerbated growth of the breast muscle [2, 3, 19]. In line with current results, access to platforms and laser lights increased locomotion and improved leg health in fast-growing broilers [19, 20, 30, 41].

Chickens chased and pecked laser lights more frequently between 1–4 weeks of age compared to older ages. Additionally, chickens used step platforms constantly during their life. This exercise may have stimulated and strengthened their locomotor system at a key timepoint of locomotor system development around 3–4 weeks of age, which is indicated by the body's peak mineral [48] and protein deposition [49] and leg growth rates [50] between 21 and 28 days of age. Thus, exercise especially at that age may have alleviated the impacts of rapid body weight gain and may have reduced postural distress, and in turn contributed to healthy vertebral bone development, reducing the prevalence of subclinical spondylolisthesis.

Access to hay bales did not impact subclinical spondylolisthesis prevalence compared to the control group. The lack of impact could be due to the type of behavior that hay bales stimulated compared to laser lights and step platform enrichments. Chickens with access to hay bales performed more pecking than chickens raised in a barren environment [30, 51], a behavior less intensive to the locomotor system compared to for instance jumping, as pecking can be performed while sitting. In addition, access to hay bales reduced locomotion compared to a barren control [51] or did not impact foraging, running, and walking [18, 52]. In this study, all these behaviors were grouped in the same category as exploratory behavior, so the effect of treatments on distinct exploratory behaviors was not tested, as this was not the study's objective. Chickens with access to hay bales explored more compared to chickens in the control and step platform treatments. The hay itself was used as a foraging substrate that was not available when housed with wooden step platforms or without enrichments. Previous studies also showed more exploration with access to hay bales compared to a control group [19, 20, 41, 51]. Additionally, our results showed that chickens rarely used hay bales for perching, possibly because it was difficult to access the top of hay bales. Thus, the low occurrence of intense exercise resulted in the lack of impact on subclinical spondylolisthesis prevalence.

Chickens with access to laser lights exhibited more frequent exploratory behavior than chickens in other treatments. Laser lights can simulate insects that elicit foraging and pecking, which are highly motivated natural behaviors also performed by domestic chicken ancestors [32, 53]. Similar to our findings, laser lights alone or combined with other enrichments increased broiler chicken activity compared to a control group [19, 32, 33]. No treatment effect was found on comfort behavior frequencies in the current study, suggesting that enrichments did not stimulate comfort behavior, while substrates such as sand do [27].

As chickens aged, they became less active, interacted with laser lights and hay bale enrichments less, and performed more comfort behaviors. Similar behavioral frequencies were

reported in previous work, where fast-growing broiler chickens reduced activity, reduced exploration, and increased comfort behaviors such as preening [19, 51, 52, 54] and dustbathing [19, 51, 52], likely due to high body weights and relatively immature locomotor systems [19, 51, 52, 54]. Bergmann et al. [44] reported lower preening and dustbathing frequencies than this study, likely due to methodological differences. They observed chickens from 01:00 to 23:00 h (rather than only during light hours), which included long periods of inactivity (during dark hours). We observed behaviors during high-activity hours only, which probably explains the higher frequencies of these comfort behaviors compared to [44]. No treatment and age interaction was found, indicating that the enrichments used in this study did not mitigate the effects of age and weight on behavior.

Performance parameters (body weight, feed intake, and feed conversion ratio) were not affected by the treatments. These results align with other studies that reported that environmental enrichments rarely alter performance [30, 36–38]. This suggests that energy requirements for increased activity levels associated with enrichments were negligible. In contrast, environments enriched with platforms, barrier perches, dustbathing areas, and wooden ramps reduced chickens'body weight gain and increased feed intake, which was theorized to be due to increased activity [55]. Similarly, other studies showed that more activity stimulated by outdoor access increased energy requirements and reduced body weight and feed efficiency [56, 57]. Yield was impacted by step platform access, with higher wing yield and lower abdominal fat in the step platform treatment than in the control treatment, but no differences between other treatments. We theorize that increased exercise throughout the rearing phase positively impacted wing yield in step platform treatment, as activities such as walking up, down, and jumping off the platforms are often accompanied by vigorous wing flapping [30], which could have increased wing muscle development and reduced abdominal fat. Exercise increases the expression of a muscle development gene (*MUSTN1*) that is part of a multi-protein transcriptional complex responsible for skeletal muscle hypertrophy regulation [58, 59]. In addition, exercise increases serum creatine kinase concentrations, which are positively associated with muscle growth rate [60, 61]. Thus, chickens in the step platform treatment may have had increased expression of *MUSTN1* and increased circulating creatine kinase concentrations compared to chickens from control treatment due to exercise, thus more wing muscle development and less abdominal fat.

The increased activity and leg exercise in laser lights and step platform treatments compared with the control group were not reflected in the chickens' gait scores. In addition, the subclinical spondylolisthesis prevalence in this study was not associated with gait scores, which is in line with expectations of a subclinical rather than a clinical diagnosis [8]. Generally, gait scores in the current study were low, reflecting good gait. Gait score is positively correlated with a range of locomotor disorders in broiler chickens, such as spondylolisthesis, tibial dyschondroplasia, valgus angular deformity, and pododermatitis [7]. The development of locomotor disorders is multifactorial, with nutritional imbalances [62–64], environmental conditions [65–67], age and stocking density [68], incubation conditions [69], and bedding material [15] impacting gait. As husbandry conditions in the current study were consistent across treatments and were meeting or exceeding commercial standards, all gait scores were better compared to broilers housed in commercial conditions [70, 71]. Thus, the prevalence of lameness was relatively low in the current study, allowing little room for improvement possible by adding enrichments. Previous studies have reported benefits of environmental enrichments on gait [18, 72, 73]. In this current study, gait worsened with age, probably due to weight gain, as fast-growing broiler chickens are prone to impaired gait as they reach slaughter weight and age [68, 70, 74].

In conclusion, these data showed that especially access to step platforms, but also to laser lights, reduced subclinical spondylolisthesis prevalence and maintained performance and yield in fast-growing broiler chickens. Furthermore, laser lights and hay bales increased exploratory behavior compared to the control group without enrichments. Gait was good, not impacted under the current study conditions, and was not associated with subclinical spondylolisthesis prevalence. Thus, our results suggest that access to environmental enrichments improved animal welfare by reducing subclinical spondylolisthesis prevalence (step platforms and laser lights) and increasing the frequencies of natural behaviors (hay bales and laser lights) without negatively impacting performance and yield. However, it is important to underline that subclinical spondylolisthesis is a multifactorial locomotor disorder and that many factors, including genetic predisposition, nutrition, environment, age, stocking density, incubation conditions, and bedding material can contribute to the etiology of spondylolisthesis [7, 11, 13–16]. Our findings suggest that environmental enrichment (step platforms and laser lights) can be used as a tool to reduce subclinical spondylolisthesis prevalence compared to a barren environment (control). These enrichments could be easily applied in commercial practice.

## Author Contributions

**Conceptualization:** Marconi Italo Lourenço da Silva, Ibiara Correia de Lima Almeida Paz.

**Data curation:** Marconi Italo Lourenço da Silva.

**Formal analysis:** Marconi Italo Lourenço da Silva, Ibiara Correia de Lima Almeida Paz, Andressa Silva Jacinto, Marcos Antonio Nascimento Filho, Ana Beatriz Santos de Oliveira, Ingrid Grazieli Althman dos Santos, Francine dos Santos Mota, Fabiana Ribeiro Caldara, Leonie Jacobs.

**Funding acquisition:** Marconi Italo Lourenço da Silva, Ibiara Correia de Lima Almeida Paz.

**Investigation:** Marconi Italo Lourenço da Silva, Andressa Silva Jacinto, Marcos Antonio Nascimento Filho, Ana Beatriz Santos de Oliveira, Ingrid Grazieli Althman dos Santos, Francine dos Santos Mota.

**Methodology:** Marconi Italo Lourenço da Silva, Ibiara Correia de Lima Almeida Paz, Andressa Silva Jacinto, Marcos Antonio Nascimento Filho, Ana Beatriz Santos de Oliveira, Ingrid Grazieli Althman dos Santos, Francine dos Santos Mota, Fabiana Ribeiro Caldara.

**Project administration:** Marconi Italo Lourenço da Silva, Ibiara Correia de Lima Almeida Paz.

**Supervision:** Ibiara Correia de Lima Almeida Paz.

**Validation:** Ibiara Correia de Lima Almeida Paz, Fabiana Ribeiro Caldara.

**Visualization:** Fabiana Ribeiro Caldara.

**Writing – original draft:** Marconi Italo Lourenço da Silva.

**Writing – review & editing:** Ibiara Correia de Lima Almeida Paz, Fabiana Ribeiro Caldara, Leonie Jacobs.

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
