## [Decision Letter · Decision Letter 0]

17 Feb 2023

PONE-D-23-01772Providing environmental enrichments can reduce subclinical spondylolisthesis prevalence without affecting performance in broiler chickensPLOS ONE

Dear Dr. Lourenço da Silva,

Thank you for submitting your manuscript to PLOS ONE. After careful consideration, we feel that it has merit but does not fully meet PLOS ONE’s publication criteria as it currently stands. Therefore, we invite you to submit a revised version of the manuscript that addresses the points raised during the review process.

We look forward to receiving your revised manuscript.

Kind regards,

Ewa Tomaszewska, DVM Ph.D

Academic Editor

PLOS ONE

Journal Requirements:

Reviewers' comments:

Reviewer's Responses to Questions

**Comments to the Author**

1. Is the manuscript technically sound, and do the data support the conclusions?

Reviewer #1: Yes

2. Has the statistical analysis been performed appropriately and rigorously? 

Reviewer #1: Yes

3. Have the authors made all data underlying the findings in their manuscript fully available?

Reviewer #1: Yes

4. Is the manuscript presented in an intelligible fashion and written in standard English?

Reviewer #1: Yes

5. Review Comments to the Author

Reviewer #1: Generally I find this experiment to be a nice and well conducted study and well presented in the manuscript. I have only minor comments and suggestions as follows:

Line 75: "When providing diets diluted in fiber in ...". Please specify type of fiber.

Line 83: Change "favor" to improve or enhance.

Line 90: Delete "beneficial" as it means the same as "enrichments".

Line 173: Change "During recording days, only daily health checks were performed to ..." to "Health checks were only performed during recording days in order to ...".

Line 190: Change "difficulty" to difficulty in".

Line 196: Please explain why the chickens were fasted for 8 hours before killing.

Line 197: You write that "The chickens were weighed and then stunned using an electric stunner (Fluxo UFX 7, Chapecó, SC, Brazil). Are you sure that the electric stunning and muscle contractions is not the reason the deforming the cervical spines and nerves? Please add a comment on this.

Line 205 and a general comment to the study: Were the same chickens observed clinically and compared to the pathology findings after killing? This could be very valuable information and also add to the strength of the posed causal relationship between pathology of the cervical spines and affected gait. Please comment on this.

Table 2: Why is performance not stratified according to week of age? This will add better information than the not mutually exclusive grouping of 1-21, 1-35 and 1-42.

Other suggestions:

- As the authors use many abbreviations, I suggest a list of abbreviations is added to the manuscript. When reading a publication, it is always very tiring to spend time on finding (re-finding) the definitions of the abbreviations after the first presentation.

- Some of the environmental enrichments have a statistically significant effect; however, the factual improvement seems to be minor. The manuscript will benefit if the authors add a comment or some estimations of the attributable effects of these environmental enrichments; I think they will be minor, and this will be a nice contribution the manuscript to underline, that this type of "problem" (spondylolisthesis) is multifactorial and that many factors each have only a small impact, and which makes it so difficult and complex to solve such problems.

6. PLOS authors have the option to publish the peer review history of their article (what does this mean?). If published, this will include your full peer review and any attached files.

Reviewer #1: No

---

## [Author Response · Author response to Decision Letter 0]

15 Mar 2023

Dear Editor,

We are very happy to receive such clear and important suggestions for improving our manuscript. We applied the corrections suggested by the reviewer, reviewed typos, clarified points that seemed unclear, and adjusted some other details as requested. The figure files were uploaded to the Preflight Analysis and Conversion Engine (PACE) digital diagnostic tool, following the journal requirements. Our detailed responses are shown in Response to reviewers file.

We look forward to receiving your decision.

---

## [Decision Letter · Decision Letter 1]

23 Mar 2023

Providing environmental enrichments can reduce subclinical spondylolisthesis prevalence without affecting performance in broiler chickens

PONE-D-23-01772R1

Dear Dr. Marconi Italo Lourenço da Silva,

We’re pleased to inform you that your manuscript has been judged scientifically suitable for publication and will be formally accepted for publication once it meets all outstanding technical requirements.

Kind regards,

Ewa Tomaszewska, DVM Ph.D

Academic Editor

PLOS ONE

Additional Editor Comments (optional):

Reviewers' comments:

Reviewer's Responses to Questions

**Comments to the Author**

1. If the authors have adequately addressed your comments raised in a previous round of review and you feel that this manuscript is now acceptable for publication, you may indicate that here to bypass the “Comments to the Author” section, enter your conflict of interest statement in the “Confidential to Editor” section, and submit your "Accept" recommendation.

Reviewer #1: All comments have been addressed

2. Is the manuscript technically sound, and do the data support the conclusions?

Reviewer #1: Yes

3. Has the statistical analysis been performed appropriately and rigorously? 

Reviewer #1: Yes

4. Have the authors made all data underlying the findings in their manuscript fully available?

Reviewer #1: Yes

5. Is the manuscript presented in an intelligible fashion and written in standard English?

Reviewer #1: Yes

6. Review Comments to the Author

Reviewer #1: Thanks for addressing all my comments and suggestions. I accept your explanation for not changing Table 2, but I still think my suggested stratification according to age, rather than the presented accumulated form, would have contributed with more knowledge, because it would have elucidated whether there is a change at a specific age interval.

7. PLOS authors have the option to publish the peer review history of their article (what does this mean?). If published, this will include your full peer review and any attached files.

Reviewer #1: No

---

## [Editor Report · Acceptance letter]

31 Mar 2023

PONE-D-23-01772R1 

Providing environmental enrichments can reduce subclinical spondylolisthesis prevalence without affecting performance in broiler chickens 

Dear Dr. Lourenço da Silva:

I'm pleased to inform you that your manuscript has been deemed suitable for publication in PLOS ONE. Congratulations! Your manuscript is now with our production department. 

Kind regards, 

on behalf of

Professor Ewa Tomaszewska 

Academic Editor

PLOS ONE